# Revolutionizing Disease Modeling: The Emergence of Organoids in Cellular Systems

**DOI:** 10.3390/cells12060930

**Published:** 2023-03-18

**Authors:** Rita Silva-Pedrosa, António José Salgado, Pedro Eduardo Ferreira

**Affiliations:** 1Life and Health Sciences Research Institute (ICVS), School of Medicine, University of Minho, Campus Gualtar, 4710-057 Braga, Portugal; asalgado@med.uminho.pt (A.J.S.); pedroferreira@med.uminho.pt (P.E.F.); 2ICVS/3B’s—PT Government Associate Laboratory, 4710-057 Braga, Portugal; 3Centre of Biological Engineering (CEB), Department of Biological Engineering, University of Minho, 4710-057 Braga, Portugal

**Keywords:** cell culture, iPSCs, disease modeling, organoids, organoids applications, organoids limitations

## Abstract

Cellular models have created opportunities to explore the characteristics of human diseases through well-established protocols, while avoiding the ethical restrictions associated with post-mortem studies and the costs associated with researching animal models. The capability of cell reprogramming, such as induced pluripotent stem cells (iPSCs) technology, solved the complications associated with human embryonic stem cells (hESC) usage. Moreover, iPSCs made significant contributions for human medicine, such as in diagnosis, therapeutic and regenerative medicine. The two-dimensional (2D) models allowed for monolayer cellular culture in vitro; however, they were surpassed by the three-dimensional (3D) cell culture system. The 3D cell culture provides higher cell–cell contact and a multi-layered cell culture, which more closely respects cellular morphology and polarity. It is more tightly able to resemble conditions in vivo and a closer approach to the architecture of human tissues, such as human organoids. Organoids are 3D cellular structures that mimic the architecture and function of native tissues. They are generated in vitro from stem cells or differentiated cells, such as epithelial or neural cells, and are used to study organ development, disease modeling, and drug discovery. Organoids have become a powerful tool for understanding the cellular and molecular mechanisms underlying human physiology, providing new insights into the pathogenesis of cancer, metabolic diseases, and brain disorders. Although organoid technology is up-and-coming, it also has some limitations that require improvements.

## 1. Introduction

Cellular disease modeling is a powerful tool used to study the underlying mechanisms of various diseases and to develop new treatments. These models can be used to study both genetic and acquired diseases, including cancer, neurodegeneration, and infectious diseases [1,2,3,4].

One of the key benefits of cellular disease modeling is that it allows the study of diseases in a controlled environment in ways that would not be possible in a living organism, such as introducing specific genetic mutations or environmental factors to study their effects on the cells. Additionally, cellular disease modeling can also be used to study complex infectious diseases that involve multiple pathways and interactions between different cell types. By studying the effects of different viruses or bacteria on the cells, it is possible uncover the underlying mechanisms of the disease and to test the effectiveness of new treatments, providing insights into the disease process [5,6,7].

The ability to obtain an embryonic-like pluripotent state from differentiated murine fibroblasts using transcription factors changed the paradigm of the stem cell biology field [8]. One of the most commonly used models for studying cellular diseases involves the use of iPSCs. Human iPSCs can be obtained from different somatic tissue cell types and can be differentiated into any specialized cell type in the human body through specific factors that control cell fate [9,10]. Additionally, iPSCs can also be used to study genetic diseases, particularly those with unfathomable genetic backgrounds, which can provide insights into the underlying mechanisms of the disease [11] or when the translation of animal models is limited [12,13,14,15,16]. Furthermore, they can serve as a relevant model for studying host species and genotype-specific parasite interactions [17,18,19,20].

Organoids technology can recapitulate the complex cellular interactions and microenvironment of native tissues, enabling the study of disease-specific mechanisms in a more physiologically relevant context than traditional 2D cell culture or animal models. For instance, organoids derived from intestinal stem cells can be used to investigate inflammatory bowel disease, while those derived from neural stem cells can be used to study neurodegenerative disorders. Furthermore, organoids can be utilized to study the effects of genetic variations or environmental factors on organ development and function, providing insights into the mechanisms that underlie human diseases [21,22,23].

## 2. iPSCs

The discovery of iPSCs from mouse ESC was first reported in 2006 by a team of scientists led by Shinya Yamanaka. They demonstrated that by introducing four specific transcription factors, octamer-binding transcription factor 3 or 4 (Oct3/4), sex-determining region Y-box 2 (SOX2), myc pronto-oncogene (c-Myc), and Kruppel-like factor 4 (Klf4), through retrovirus-mediated transfection, they could reprogram adult cells into a pluripotent state. In 2007, the same team generated and cultured iPSCs from adult human fibroblasts using the same retroviral transduction factors, which exhibited similar characteristics and differentiation potential in vitro and teratomas to hESC [9]. Regarding the reprogramming process, iPSCs exhibit similar results in morphology, proliferation, gene expression, and teratomas compared to embryonic stem cells, avoiding the ethical concerns related with the use of embryonic stem cells [8,24,25,26]. This breakthrough was a significant step forward in the field of stem cell research, as it provided a way to generate patient-specific pluripotent cells without the need for embryos, which had been a major ethical concern in the field [26,27].

Since the discovery of iPSCs, there have been many advancements in the technology used to generate and manipulate these cells. The use of viral vectors to introduce the reprogramming factors was replaced by non-viral methods, such as the use of plasmids or proteins, which eliminated the risk of insertional mutagenesis. Additionally, the number of reprogramming factors required were reduced, making the process more efficient and cost-effective [28,29]. Some of the advantages of using iPSCs include:Ethical concerns: iPSCs can be derived from adult cells, avoiding the ethical concerns associated with embryonic stem cell research [30];Patient-specific cells: iPSCs can be generated from a patient’s own cells, allowing for the creation of patient-specific cells for use in therapy. This can help to avoid immune rejection of the transplanted cells [31];Disease modeling: iPSCs can be used to create models of specific diseases, which can aid in the understanding of the disease and the development of new treatments [18];Drug development: iPSCs can be used to test the safety and efficacy of new drugs on a variety of cell types, including those that are difficult to obtain from living donors [32,33];Tissue repair and regeneration: iPSCs have the potential to be used to repair or regenerate damaged or diseased tissue, such as in the treatment of heart disease, diabetes, and neurodegenerative disorders [15];Cost-effective: iPSCs can be created from a small sample of adult cells; however, can be very cost-effective compare to other methods [34,35,36].

The potential applications of iPSCs are vast, and ongoing research is being conducted in several areas. One of the most promising fields is regenerative medicine, where iPSCs can be used to generate specific cell types for tissue engineering and cell replacement therapy. For example, iPSCs can be differentiated into heart muscle cells, which can then be used to repair damaged heart tissue in patients with heart disease [37,38,39]. Similarly, iPSCs can be differentiated into insulin-producing cells, which can be used to treat patients with diabetes [40,41].

iPSCs differentiated into hepatocyte-like cells can be a good substitute for primary human hepatocytes because they can be maintained in long-term culture. Moreover, generating cells from patients with specific genetic backgrounds to study genotype-phenotype relationships may in the future avoid the need for organ donation [42,43]. 

Proof-of-concept studies revealed the efficacy of these cells in drug development, high-throughput drug screening, and modeling liver diseases and viral and parasitic infections, such as malaria [44,45]. 

Another area of research for iPSCs is drug discovery and development. Through the generation of iPSCs from patients with specific diseases, in vitro models of these diseases can be created and used to study the underlying causes of the disease and to test the efficacy of new drugs. This can greatly accelerate the drug development process and ultimately lead to new treatments for patients [46,47].

iPSCs also have potential applications in personalized medicine, with iPSCs generated from a patient’s own cells; researchers can create patient-specific cells that can be used in cell-based therapies. Additionally, iPSCs can be used to generate patient-specific organoids, which are miniaturized versions of organs that can be used to study the development and progression of diseases [48,49].

iPSCs technology was extensively studied for neurological disorders in recent years due to the limitations of studying the human brain. iPSCs can be directly generated from patients with neurological diseases, allowing the study of nervous system diseases in vitro using 2D, 3D, and BBB models [50,51,52,53]. The iPSCs differentiation and their use as in vitro model of neurodegenerative disorders, such as Parkinson’s disease (PD), enables a better understanding of neuronal cell death and contributes to the development of drug discovery [47,54]. iPSC derived from PD and differentiated into dopaminergic neurons revealed disease-related cell phenotypes, such as impaired mitochondrial function, increased oxidative stress, increased apoptosis, and an accumulation of α-synuclein, providing more insights into PD pathophysiology [51,55].

The iPSC lines derived from Alzheimer’s disease (AD) patients, a neurodegenerative disease characterized by β-amyloid (Aβ) plaques and neurofibrillary tangles with progressive cognitive decline, showed neurons differentiated with disease characteristics. A study demonstrated an increased Aβ42:40 percentage of generated cortical neurons from AD patients-iPSC lines compared with control neurons of non-AD iPSCs and different signatures for Aβ43, Aβ38, Aβ fragments in 2D and 3D cultures [56]. Hence, iPSCs from patient-derived neural cells can reflect the human genetics and physiology of AD and other dementia diseases, being an excellent cellular model to explore Alzheimer’s pathogenesis and lead to neuronal drug discovery [57,58].

However, there are also some ethical concerns surrounding the use of iPSCs. One concern is that the use of iPSCs in research and therapy could perpetuate inequalities in access to healthcare, as the technology to generate and manipulate these cells is currently expensive and not widely available [59]. Moreover, there are concerns about the limitation of genetic diversity, which jeopardizes equity and can impede the acceleration of biological discoveries [60]. There is also potential for the use of iPSCs in controversial areas, such as human cloning or the creation of genetically modified organisms [61,62].

The iPSCs technology also has several technical and scientific drawbacks that must be overcome for its full potential to be realized. One major technical weakness of iPSC technology is the variability in the quality and characteristics of iPSCs generated from different sources. iPSCs can vary in their differentiation potential, genomic stability, epigenetic modifications, and gene expression patterns. These variations can be influenced by the methods of reprogramming, the quality of the starting cells, and the culture conditions used to maintain the iPSCs. This variability can make it difficult to compare results between studies and to predict the behavior of iPSC-derived cells in a clinical setting [63,64,65,66]. 

The efficiency of the iPSCs reprogramming process can be low, especially for certain cell types or patient populations. Additionally, the reprogramming process can be time-consuming and expensive, requiring the use of multiple factors and genetic modifications. Scientifically, one of the major drawbacks of iPSC technology is the potential for genetic abnormalities to be introduced during the reprogramming process. Reprogramming can lead to changes in gene expression, DNA methylation patterns, and chromosomal stability, which can alter the properties and behavior of iPSCs and their differentiated derivatives. These changes can also increase the risk of tumor formation or other adverse effects when iPSC-derived cells are used in a clinical context [67,68,69].

Furthermore, iPSC technology faces challenges in differentiating into specific cell types with high efficiency and fidelity. The protocols for differentiation can be complex and require optimization for each cell type, and there can be significant variability in the quality and purity of the differentiated cells produced. Additionally, there may be functional differences between iPSC-derived cells and their natural counterparts, which may limit their therapeutic potential [70,71,72,73,74]. 

In conclusion, due to the plasticity of iPSCs in differentiation into several cell types and their self-renewal capacity, they constitute a vital tool for research in regenerative medicine, neurological disease modeling, cell therapies, and drug screening [12,47,54,57,75,76,77,78,79,80,81,82]. However, iPSC technology holds technical and scientific hurdles that must be overcome to fully realize its potential. Continued research and development in iPSC technology will be critical to address these challenges and to enable the safe and effective use of iPSC-derived cells in clinical applications, as well as to address the ethical concerns surrounding their use [30].

## 3. Cell Culture System—From 2D to 3D 

Cell culture is the process of growing cells in a controlled environment outside of their natural setting; the two main types of cell culture systems are 2D and 3D [83].

The 2D cell culture system refers to the growth of cells on a flat surface, such as a Petri dish, microscope slide, or a culture plate. This is the most common and well-established type of cell culture, and it is widely used in basic research, drug discovery, and biotechnology [84]. In a 2D cell culture system, cells are grown in monolayers and are typically maintained in a liquid medium, such as nutrient broth or serum-containing medium, which provides the necessary nutrients and growth factors for cell growth and proliferation [84,85,86,87]. 

An important application of 2D culture is the production of biological products. Many of these products, such as monoclonal antibodies and recombinant proteins, are produced through cells that were grown in 2D culture. These cells can be genetically engineered to produce specific proteins, which are then harvested and purified to create the final product [88,89].

The usage of iPSCs in 2D culture can help better understand neurotransmission, the central nervous system, and the differentiation of neurons, astrocytes, and microglia [90], giving more insights into the genetic and molecular conditions of neurological disorders [76]. 

Advantages of 2D cell culture include its simplicity and ease of use, as well as its accessibility and low cost, due to the easy set up and maintenance, the cells can be easily observed and manipulated. Additionally, 2D cell culture is well-suited for growing a wide range of cell types, including primary cells, stem cells, and cancer cells and allows the differentiation of specific subtypes of cells in a dish [91,92,93]. 

Although 2D cell culture is widely used and valuable tool in laboratory research, it has a limited role in the disease modeling because it cannot accurately resemble the dynamic complexity of the in vivo environment, cell–cell communications, and tissue- and organ-level structures. This is because 2D cultures can only do so for differentiation of one cell type in a mono-culture system [76,90,94,95,96,97], leading to artificial behavior of cells, such as altered proliferation and differentiation, and reduced cellular interaction compared to in vivo conditions [98,99]. For example, drug development has a high-cost when the process goes from target identification stage to clinical use. The drug screening performed in cells cultured in 2D is not representative of cells in a tissue microenvironment, resulting in a high failure rate in drug discovery and low levels of approved drugs in the market [100].

Despite these limitations, 2D cell culture remains a valuable tool in the laboratory, and recent advances improved the system, such as techniques for analyzing and characterizing cells, such as high-content imaging and transcriptomics.

The use of 3D cell culture systems started more than a decade ago and allow more complexity between cells’ interaction with heterotypic settings, mimicking multiple structures, using scaffold-based or scaffold-free cultures [76,101]. This system present similarities with tissue architecture in vivo, respecting the phenotypic and functional characteristics and circumventing a limitation of monolayer cultures [102,103]. 

Despite these limitations, 2D cell culture remains a valuable tool in the laboratory, and many advances were made in recent years to improve the system, such as techniques for analyzing and characterizing cells, such as high-content imaging and transcriptomics. These techniques allow for a more detailed understanding of the behavior and characteristics of cells in culture [104,105]. 

Scaffolds-based techniques are typically made of synthetic or natural materials, such as hydrogel-based supports, which provide a physical structure for cells to attach and grow on. Scaffold-free technique can be used to create small structures, such as microchannels and microwells, that can control cell behavior, such as suspended microplates or spheroids with coated ultra-low attachment microplates. Hydrogels are hydrophilic polymeric materials that can mimic the extracellular matrix (ECM), such as fibrinogen, hyaluronic acid, collagen, matrigel, or gelatin, providing a supportive environment for cells. They allow soluble factors such as cytokines and growth factors to move through the tissue-like gel [87,100]. 

The 3D cell culture system also allows for the study of cell–cell interactions, cell-matrix interactions, and cell-microenvironment interactions that closely mimics the in vivo microenvironment [51,106]. Moreover, 3D cell culture enables the formation of complex cellular structures, such as spheroids involved in bone regeneration [107] and organoids. These features provide insights about the organ’s behavior and bridge the gap between 2D cell culture and animal models [87]. 

Organoids and spheroids are 3D structures that are increasingly used in biological and medical research, as they better mimic the in vivo environment compared to traditional 2D cell cultures. Although organoids and spheroids have some similarities, they have distinct differences that distinguish them from one another [108].

Organoids are 3D structures derived from stem cells or tissue explants that can self-organize into structures resembling specific organs. They exhibit complex cellular organization, spatial orientation, and function similar to that of the in vivo tissue, composed of multiple cell types, and can be used to study organ development, disease modeling, and drug screening. They are typically cultured in a specialized medium containing growth factors that promote differentiation and tissue-specific gene expression [108,109,110,111]. Spheroids, on the other hand, are aggregates of cells that form a 3D structure, but lack the complex organization and functional specialization of organoids, and often have a homogenous cell population. They can be generated by culturing cells in non-adherent conditions, such as suspension cultures, hanging drops, or microfluidic devices. Spheroids can be composed of a single cell type or multiple cell types and can be used to study cell–cell interactions, drug screening, and tumor biology [112,113,114,115].

Other 3D structures that can be generated in vitro include organotypic cultures, tissue-engineered constructs, and scaffolds. Organotypic cultures are 3D cultures that resemble the in vivo tissue architecture but lack the self-organization seen in organoids. Tissue-engineered constructs are artificial structures that are created using cells and biomaterials and can be used for tissue engineering applications. Scaffolds are 3D structures that provide support for cells to grow and differentiate, and can be used for tissue engineering, drug screening, and regenerative medicine applications [116,117,118,119,120]. In summary, while organoids, spheroids, and other 3D structures have some similarities, they can be distinguished by their cellular organization, complexity, and functional specialization. Organoids are self-organized structures that resemble specific organs, while spheroids are simple aggregates of cells lacking distinct organization. Other 3D structures, such as organotypic cultures, tissue-engineered constructs, and scaffolds, are useful for a range of applications, including tissue engineering and drug screening.

Additionally, the 3D cell culture has the potential to provide different cell types, giving helpful information about tumor cell biology, signal transduction, cell migration, drug discovery, angiogenesis, metabolic profiling, inflammation, and apoptosis. Moreover, the 3D culture model exhibits cellular behavior that is more similar to in vivo in responses to oxidative stress compared to the 2D model [121]. For example, the 3D culture allowed us to understand how the formative components of the cerebellar structure interact and how they self-organize and differentiate into Purkinje cells. The addition of the factors lead to a polarized design with a rhombic-lip-like layout and a three-layered cytoarchitecture similar to the human embryonic cerebellum, as in the first trimester [122]. The 3D model applied in neural cell culture of AD pathogenesis study showed relevant information about amyloid-β (Aβ) aggregation and increased concentration of hyperphosphorylated tau [123,124].

The 3D model could be used for pre-clinical studies at a low cost, such as for the gastrointestinal tract–liver system [125]. It is a potential new model for pre-clinical stage research, providing relevant information about other types of diseases, and could contribute to the use of fewer animal models [87,125,126]. Moreover, it is a model that can be used to study drug efficacy and toxicity because it can provide a more realistic model for drug testing, allowing for a better understanding of how drugs interact with cells. However, there are also some limitations associated with 3D cell culture, such as the complexity of the system and the difficulty of monitoring the cells within it. With the growing interest in 3D cell culture, it is expected that this field will continue to develop and provide new opportunities for research and drug development in the future [87,100].

In conclusion, 2D and 3D cell culture systems are both important tools in cell biology research and biotechnology, but they have distinct advantages and limitations. The 2D cell culture system is well-established, simple, and accessible, while 3D cell culture more closely mimics the in vivo environment of cells, making it useful for studying certain types of biological processes. Both 2D and 3D cell culture are important for understanding the biology of cells and for developing new therapies [99,127].

## 4. Organoids Technology

Organoids are 3D structures, miniaturized versions of organs that are grown in a lab using stem cells or other cell types. These structures closely mimic the architecture, function, and behavior of their in vivo counterparts, making them valuable tools for studying a wide range of biological processes and diseases [111,128].

Organoids can form endodermal, mesodermal, and ectodermal organs. Organs derived from cells of the endoderm layer are usually associated with complex systems, including the gastrointestinal and respiratory tract, and all their associated organs. Mesoderm derivatives are associated with blood vessels, muscles, kidneys, heart, bone, cartilage, and reproductive organs. Those derived from the ectoderm, from the cells of the epithelium layers, are related to the nervous system, such as the brain [22]. Additionally, organoids can be established in different ways, through the extracellular matrix, such as collagen or matrigel, and with differentiating factors, in plates, or rotating bioreactors, which are usually used for cerebral organoids as they allow greater medium perfusion. Air-liquid interface (ALI) can also generate organoids, such as microfluidic, in which the top layer of cells is exposed to air, and the basal layer is in contact with the culture medium, usually applied to renal, gastrointestinal, and neural organoids [22,129,130,131].

Organoids have gained particular focus as an appealing model since they are 3D self-organized structures and the microenvironment is preserved, including morphological and biological issues of organs. They can mimic genetic diseases, host-infectious disease interaction, regenerative therapy, or drug screening using patient-specific iPSCs cells and establishing biobanks. Furthermore, through molecular technologies, such as the lentiviral expression system and CRISPR/CAS9, it is possible to manipulate the genome of organoids, allowing for disease replication and targeted gene therapy [23,76,132,133] (Figure 1).

Organoids can be used to study human biology and disease in a way that is not possible with traditional cell culture or animal models. For example, organoids can be derived from patient-specific stem cells, allowing researchers to study the effects of genetic mutations on organ development and function in a way that is not feasible in vivo. Organoids have been used to study a wide range of organs and systems, including the brain, liver, pancreas, and gut. For instance, brain organoids were used to study the development of neural circuits and the effects of genetic mutations on brain function [134,135,136,137]. 

Organoids are a potential tool to study organ transplantation and the process of transplant rejection, as well as to test new drugs and other treatments that could be used to prevent rejection. Understanding genetic mutations associated with organ-specific diseases, as well as how genetic mutations affect organ development and function, are important factors that can be explored [138,139,140].

Organoids are a powerful tool for studying human biology and disease, as they allow for the study of the effects of genetic mutations and environmental factors on organ development and function, which is not possible with traditional cell culture or animal models. In addition, organoids can be used to study the efficacy and toxicity of different drug treatments, providing a more physiologically relevant context for drug screening or pathways, providing insights into the mechanisms of drug action. As the technology continues to improve, organoids are likely to play an increasingly important role in scientific research and the development of new treatments for a wide range of diseases. Moreover, organoids have the potential to revolutionize the way of study human diseases and develop new therapeutics, but further research is needed to fully realize their potential. With the continued growth of organoid technology, significant advances are expected in the fields of disease modeling, drug discovery, and personalized medicine [141].

## 5. Organoids Modeling Neurological Diseases

An in vitro neurological organogenesis model can be created, using iPSCs to generate a human cerebral organoid [79,106,142,143,144]. The culture of cerebral organoids involves using cell differentiation factors and a spinning bioreactor to improve media perfusion. These organoids resemble the multilayered structure and brain cell types found in the human brain, such as astrocytes, oligodendrocytes, neuroepithelial cells, neural rosettes with well-defined apical-basal polarity, and epithelial features similar to the embryonic neural tube, neuronal cells, and radial glial cells [21,145]. Specific layers within the cortical plate (CP), such as the ventricular zone (VZ), subventricular zone (SVZ), and intermediate zone (IZ), can be created (Figure 2). Additionally, it is possible to generate specific brain regions in cerebral organoids, such as the forebrain, midbrain, hippocampus, prefrontal cortex, cerebellum, and occipital lobe, which allows for the study and modeling of neurological disorders [7,146]. The neurons in cerebral organoids exhibit synapse structures and are active and responsive with calcium action potentials, which can be blocked using biochemical product [143,144,147]. Moreover, a cerebral organoid can be composed of multiple cells and resemble the fetal cerebral with structured cerebral regions, such as the cerebral cortex, hindbrain/midbrain, and hypothalamus [53,101,147,148,149]. Furthermore, cerebral organoids can be composed of different cell types, such as neural stem cells, mature and immature neurons, or glial cells, with neural connectivity similar to the human brain in vivo [145]. These features enable the study of structural phenotypes and a deeper understanding of cellular and molecular mechanisms related to human brain development, disease-induced neuronal illnesses, and cognitive impairments [150]. However, understanding the neuropathogenesis of infectious parasitic diseases, including cerebral malaria [151,152] and toxoplasmosis [153] arepossible to achieve through manipulation and modeling analysis [52,76,154,155,156]. Forebrain-specific organoids exhibit progenitor zone organization, neurogenesis, gene expression, and a human-specific outer radial glia cell layer, which are the main traits of human cortical development [7]. Midbrain organoids contain dopaminergic neurons, astrocytes, and oligodendrocytes [157], while hindbrain [158] and thalamus-like [159] organoids can also be generated. Single-cell RNA sequencing (scRNA-seq) revealed that the cortex-like regions of cerebral organoids are similarly organized and structured to the fetal neocortex, using homogeneous gene expression programs [160]. A recent study involving the co-culture of HUVEC spheroids with a vascularized system developed together with cerebral organoids in vitro induced angiogenesis in the cerebral organoid, which guides the Wnt/β-catenin pathway expression. The vascularization of cerebral organoids resulted in the promotion of neural proliferation and differentiation, reduced apoptosis, the inhibition of necrosis, and activation of signaling pathways related to cerebral organoid development [161]. Cerebral organoids are a versatile model system used to understand diseases, such as Zika-associated microcephaly (ZIKV) [7,162,163,164] that induce microcephaly, Alzheimer’s disease (AD) [165], autism-spectrum disorder, and glioblastoma multiforme Research studies have been conducted with the aim of advancing the understanding and application of personalized medicine, investigating new testing methods for therapeutics [166], as well as deepening knowledge of neurological disorders and brain cancers [167]. 

Therefore, brain organoids have the ability to form complex neural networks and exhibit spontaneous electrical activity, making them a valuable tool for studying the underlying mechanisms of these disorders. Although brain organoid technology has limitations, they can become a robust system to preform biopharmaceutical drugs screening and neurotoxicology studies [168,169,170].

## 6. Organoids Modeling Infectious Diseases

One of the most significant applications of organoids in infectious diseases is their potential for studying the pathogenesis of infections, creating a dynamic microenvironment closer to reality, and mocking tissue structure. Moreover, being useful for understanding cellular and molecular mechanisms involved in infection host–pathogen interactions and the response of the host to the infection [171,172,173,174,175]. 

Since organoids can be created with specific patient characteristics, they can mimic the outcome of host–pathogen interactions as an advanced model. Organoids were used as models of infections caused by viruses and to study the pathogenesis of respiratory infections, such as severe acute respiratory syndrome coronavirus 2 (SARS-CoV-2), or microcephaly as ZIKV. Moreover, they can also be used to study the inflammatory response of the host to the infection and the recruitment of immune cells, such as in the human immunodeficiency virus (HIV). This can provide insight into the mechanisms underlying the disease and the host’s response to it [176,177,178,179]. 

The use of brain organoids made it possible to identify the neurotropism of SARS-CoV-2 for epithelial cells of the choroid plexus, damaging this barrier, which protects the entry of pathogens into the brain. In addition, infected cells expressed apolipoprotein and the viral receptor angiotensinogen 2 [177,179,180]. Organoid exposure to SARS-CoV-2 revealed metabolic alterations [181], aberrant Tau localization in neurons, specific targeting of the cortical region, hyperphosphorylation, neuronal death [182], and transcriptional dysregulation [177].

Brain organoids recapitulated microcephaly disease [147] and, when infected with ZIKV, it induced Toll-like-Receptor 3 activation, a decrease in organoid size, resembling microcephaly, which, in turn, is linked to the number of viral copies [162,164]. Brazilian ZIKV-infected human brain organoids showed reduction in proliferative zones and disrupted cortical layers [183]. The human iPSC-derived forebrain organoid modeled ZIKV exposure and exhibited increased cell death and reduced cell proliferation, with a consequent decrease in the volume of the neuronal cell layer similar to microcephaly [7]. Another study with cerebral organoids showed that the ZIKV impairs cortical growth and folding. Therefore, this indicates that cerebral or region-specific organoids can model human brain development and disease or its potential for drug testing [181]. 

Herpes simplex virus can induce impairments in central nervous system (CNS), especially in newborns, with consequent long-term sequelae of patients and mortality [184]. When used to infect brain organoids, the organoids showed alterations in the growth, morphology, neuroepithelial identity, and transcriptional signatures [185]. Moreover, cerebral organoids showed that the virus can move from the periphery to the central layers [186]. 

Brain organoids can model aspects of transmissible human prion disease, the most common being the Creutzfeldt–Jakob disease. Prion disease is responsible for inducing severe neurodegeneration and it is fatal [187,188]. 

The HIV induces changes in the CNS, allocating itself in astrocytes, contributing to the HIV burden and neurological dysfunctions. Microglia-containing human brain organoids showed infection by HIV through the CCR5 and CD4 co-receptor, and productive HIV infection was observed, with microglia being the target cell of HIV [189] and releasing tumor necrosis factor and interleukin-1β, and resembled the neuroinflammatory environment [178]. 

Congenital human cytomegalovirus (HCMV) infection causes central and peripheral nervous system disorders as well as microcephaly. HCMV-infected brain organoids demonstrated neurodevelopmental changes in organoids, a reduction in cortical structures, changes in organization and depth of lamination in these structures, and aberrant expression of the neural marker β-tubulin III [190]. Cerebral organoids are a model of HCMV showing alterations in calcium signaling, abnormal neural activity, reduced growth, and impaired cortical layers [191]

The Japanese encephalitis virus (JEV) affects brain cells and causes the disease Japanese encephalitis, which has no cure. Telencephalon organoids injected with JEV showed cell shrinkage, cell death, and smaller organoids in size. Furthermore, a preferential infection of astrocytes and neural progenitors, such as external radial glial cells were found. At different stages of culture of the organoids (above 8 weeks), the activation of the interferon signaling pathway occurred [192].

Brain organoids can be disease models for West Nile virus, dengue virus, and varicella-zoster virus as they induce neurological diseases. Differentiated human neural stem cells [193] and iPSCs [194] were used to study these viruses, demonstrating the usefulness of infection models. Thus, the use of brain organoids becomes quite promising as an infection model of neurodevelopmental and neurodegenerative diseases [195,196].

Gastrointestinal organoids have an epithelial layer around the central lumen, apical side, and cell polarization. They may also have structures similar to intestinal crypts or gastric glands and present specific types of cells [197]. Organoids were repeatedly used to provide scientific advances of diseases, such as human fallopian tube organoids to study infections caused by *Chlamydia trachomatis* [198] and intestinal organoids to model *Escherichia coli* intestinal infection [199].

There are different techniques for exposing organoids to infection. It can be through the microinjection of infectious agents, such as *Helicobacter pylori*, into the lumen of gastric organoids. Although there is a similar pathogenic response as with in vivo tissue, this technology has some limitations, such as the lack of control in the number of infections, due to the different numbers and types of cells per organoid, and the variability in the diameter of the organoids limiting the standardization of the infection [197,200]. Infection of organoids during cell passage is also one of the methods used. Organoids are crushed into fragments and incubated with pathogens, and are then incorporated into the extracellular matrix to generate a new organoid [197].

As a result, organoids are also useful for drug discovery in infectious diseases. They can be used to screen drugs for their effectiveness and toxicity in a relevant tissue context. For example, organoids derived from the lung can be used to screen drugs for their effectiveness against respiratory infections, such as influenza, while organoids derived from the gut can be used to screen drugs for their effectiveness against enteric infections, such as *Bacillus subtilis* [201], *Salmonella* [202], *Listeria* [203], or *Shigella* [204]. Organoids can also be used to study the mechanisms of drug action and the resistance mechanisms of the pathogen. This can provide insight into the design of new drugs that are more effective and have fewer side effects [205].

## 7. Organoids Modeling Several Diseases 

One of the most promising areas of organoids research is in cancer to study the development and progression of different types of cancer, the tumorigenesis mechanisms as well as the genetic and epigenetic changes that drive tumor development. Cancer organoids were generated from patient-derived tumor samples, which can be used to study the genetic and molecular changes that occur during tumor progression; for example, to test the effectiveness of different cancer drugs or to investigate the mechanisms that drive tumor growth and metastasis. Moreover, they can induce specific genetic alterations in cancer organoids using various techniques, such as gene editing. The cancer organoids are then analyzed to assess the effects of induced changes on tumor progression, which includes analysis of cell proliferation, migration, and invasion, as well as changes in gene expression and protein activity [206,207].

Additionally, organoids can be used to study the response of cancer cells to different therapies, such as radiation and chemotherapy, providing insights into drug resistance mechanisms and the identification of new therapeutic targets [208,209,210,211].

Intestinal organoids showed the presence of different types of differentiated cells, different intestinal crypts, and villi [212,213,214], and the effects of intestinal inflammation and infection on gut were studied [215,216,217]. The liver organoids were also developed as a model system that recapitulate the heterogeneity of patients liver cancer, in phenotype, cancer cell composition, and treatment reaction [218]. The liver organoids were also used to study the effects of toxins, discovery of potential new therapies, and other environmental factors on liver function [219,220]. 

The organoid model of pancreatic cancer histologically resembles human pancreatic cancer [221] with identification of pathways involved in oncology therapy [222]. Moreover, pancreas organoids were used to study diabetes and other metabolic disorders [223,224].

Endometrial cancer-derived organoids replicate the diversity of endometrial diseases and reproduce the original lesion after transplantation in vivo [225]. Moreover, endometrial organoids were used to study infertility [226]. These organoids can be used to study the underlying mechanisms of these diseases, as well as to test the efficacy of different drug treatments [227]. 

Organoids have the potential to make a significant impact in the study of developmental biology. Organoids can be used to study the development of different organs and to understand how they form and function. This can help researchers understand the underlying causes of congenital defects and can also be used to test the effectiveness of new treatments for these conditions. Human cardiac organoid shows similar constructs to in vivo organ, modeling cardiovascular disorders, such as heart failure, genetic cardiac diseases, and arrhythmia, providing a valuable drug testing platform [228,229,230,231]. Additionally, organoids were used to study immune tissue structures and functions, such as gut organoids for bowel disease [232], thymus organoids to induce donor-specific immune tolerance to allografts [233], lymph node organoids for regulating antibody production [234], and intestinal organoids to study Crohn’s disease [235].

Furthermore, organoids can also be used in drug discovery and development. They can use organoids to study the interactions of drugs with specific cell types and organs and can help understand the efficacy and side-effects of drugs, developing better and more targeted drugs. These models can be used to study the behavior of cancer cells and to test the effectiveness of different treatments. Additionally, organoids can be used to study the interactions between cancer cells and the surrounding microenvironment, which is crucial for understanding how cancer progresses and how it can be treated [236,237,238].

In drug toxicology studies, organoids are a new disease model for tumors, hereditary, infectious, neuropsychiatric, and neurological diseases. Although organoids have limitations, it is a model adopted to mimic various organs and pathologies as well as for drug testing. Nevertheless, improvements are required to make organoids a model with additional applicability in the clinical treatment of patients [239].

## 8. Future Perspective Application of Organoid Technology in Research

Organoid technology is a rapidly developing field with huge potential for research in various areas and it is likely that it will continue to be an important tool for a wide range of fields. From cancer research to neuroscience, developmental biology, gut-microbiome interactions, and drug discovery, organoids can provide new insights and understanding into a wide range of biological processes and diseases. Future efforts in organoid technology will be key to investigate more complex interactions in human organs. For example, it will allow a more comprehensive range of disorders at different stages of human development and modeling tissues (Figure 3). As a result, diagnostic and therapeutic strategies can be improved, and therapeutic screening at a higher performance level [22].

## 9. Regenerative Medicine

Regenerative medicine is based on replacing diseased tissue with healthy tissue through allogeneic transplantation. However, donor compatibility, difficulty obtaining healthy tissue, and immune rejection are some severe limitations to this type of therapy. Thus, in recent times, regenerative medicine focuses on stem cell research, tissue engineering, and patient-derived models. Tissue engineering is the process of creating functional tissue by combining cells, biomaterials, and growth factors in a controlled environment [240,241]. 

Organoid technology is a rapidly advancing field and raises expectations in regenerative medicine that has the potential to revolutionize the way we approach tissue repair, regeneration, and transplantation. It provides a source of cells that can be used to regenerate damaged or diseased tissue without the need for a donor organ. Moreover, organoid technology allows the development of isogenic organoids from patient tissues and may serve as an alternative method to organ replacement strategies [242,243,244]. For example, organoids derived from stem cells can be used to create functional liver tissue, which can be used to treat liver diseases such as cirrhosis and liver cancer [245,246,247,248].

There is a growing interest in using intestinal epithelial cells in vitro to form intestinal organoids as a source for tissue engineering and regenerative medicine. It showed that intestinal stem cells can be grown in vitro as organoids and that they can be used for transplantation studies. Furthermore, studies in mouse colitis models showed that ex vivo cultured organoids can be grafted into colitis ulcers, rebuilding the crypt-villi structures, and decreasing the risk of colitis-associated cancers. Organoid transplantation could become a future therapy for patients with refractory inflammatory bowel disease [249,250]. Kidney organoids were also the focus of increased attention because the kidney is a complex organ with many types of cells. Kidney organoids showed great potential for understanding renal pathophysiology when applied to renal injury and fibrosis, despite the reproducibility difficulties. It is expected that with the evolution of their maturation and functionality, they may later be used as a renal replacement in regenerative medicine [251,252].

The variability of organoids depending on the culture protocol and the biocompatibility of an extracellular matrix for cell transplant therapy are vital points to be improved in organoids for their application in the future of regenerative medicine and personalized medicine. Furthermore, the safety of the entire process from the production of cells to validation as a model needs to be improved, and in addition, the cost of manufacture and confirmation of the effectiveness of the therapy in vivo must be established [253].

One of the major challenges in using organoids for transplantation is the lack of immune compatibility. Organoids derived from stem cells are not immune-matched to the patient, which can lead to rejection of the transplant. However, recent advances in gene editing techniques, such as CRISPR-Cas9, made it possible to genetically modify organoids to make them immune-compatible with the patient. This is a major step forward in the development of organoid-based therapies and has the potential to greatly increase the number of patients who can benefit from organoid transplantation [254,255].

## 10. Bioengineered Organoids and 3D Bioprinting

Bioengineered organoids and 3D bioprinting revolutionized the field of tissue engineering and regenerative medicine, allowing for the creation and induction in molecular, cellular, and structural characteristics relevant in vitro models of human organs [256]. 

It is promising that tissue and organ engineering enable patterning human tissues composed of different cell types with a similar physiological microenvironment that can be maintained over prolonged periods. Moreover, this might contribute to developing organoids with greater homogeneity and vascularization in the future [257,258,259].

The combined bioengineering with self-organizing organoid properties of cells can be used to create bioengineered organoids with improved reproducibility and tissue structure. For example, the poly lactide-co-glycolide copolymer (PLGA) fiber microfilaments can be utilized as a floating scaffold, obtaining elongated EBs. As a result, the cerebral organoids showed a well-defined neuroectoderm formation and cortical development, including spatial organization of polarized cortical plate and radial elements [260].

The 3D bioprinting is a technology that allows the construction of complex models of tissues and organs with different types of cells. It is possible to build models with varying combinations of materials, bioactive molecules, and printing methods involved in functional 3D structures. These structures include organoids, tissue constructs, and even functional organs. Additionally, it is possible to obtain models with high accuracy, functionality, repeatability, and reproducibility. With 3D bioprinting technology, it is possible to precisely define the external and internal geometry, spatial organization, and cellular orientation of the formed tissues to mimic the structure and function of their biological peers. Moreover, it allows interconnectivity of different regions within the organ and the correct perfusion of the nutrients for proper tissue development or repair [261] or can generate vascular networks in organoids [259]. 

Several bioprinting methods include extrusion, inkjet, laser, dual head printing, and light-mediated stereolithography [261,262]. The 3D bioprinting process is composed of cells printed in basic matrix bioinks, such as hydrogels. Bioinks are printable biomaterials used in 3D printing that are important for precision medicine due to the use of tissue organoids with diverse applications, such as drug screening (testing the efficacy and toxicity of drugs) or patient-derived xenografts [263]. Hence, the 3D bioprinting can be used to develop more physiologically relevant in vitro models for drug discovery and development [264,265].

In brief, the 3D bioprinting may impact clinical limitations, with a significant impact on regenerative medicine and organoid technology, creating the 3D bioprinting of organoids, such as improving the formation of bones and muscles. In addition, the 3D bioprinting of organoids may contribute to studying tumorigenesis, microenvironmental specificities, the cellular and functional changes of the cerebral cortex, or accelerating the treatment of patients with tissue or organ disabilities, cancer, or aging issues [266,267,268].

## 11. Gene Editing Technologies in Organoids 

Organoids are a powerful tool to increase knowledge about different types of cells, cell characterization, and discovery of the mechanisms and signaling pathways involved in diseases when combined with a transcriptomic approach, such as single-cell RNA sequencing (RNA-Seq). Furthermore, gene editing technologies, such as CRISPR-Cas9, revolutionized the field of biology by allowing researchers to make targeted changes to the genome. With genome editing techniques, it is possible to mimic with organoids the genetic mutations that occur in diseases of human patients, to understand more deeply the pathologies and the function of genes. Thus, molecular biology techniques not only help to find new biomarkers, but the organoids derived from patients can also be genetically repaired so that a personalized treatment strategy can be established [151,256].

With CRISPR-CAS9 technology, gene editing has become simpler, allowing researchers to introduce specific mutations and study their effects on organ development and function. Furthermore, CRISPR-CAS9, in the case of tumors, enables the production of specific subtypes of tumor organoids, evaluating the carcinogenic activity and assessing cancer incidence, cancer progression, and cancer metastasis. For example, in prostate cancer patients, the BRCA2-RB1 deletion was found to alter epithelial-stromal and increase cancer aggressiveness. In brain cancers, with gene editing and the use of neoplastic cerebral organoids, combinations of mutations that lead to neuroectodermal tumors were identified [132,257]. Moreover, CRISPR–Cas9 is used as reverse-engineering to understand the serrated neoplasia pathway in organoid-based colorectal cancer [258]. Usage of CRISPR-Cas9 involves screening human colon organoids for patient-specific functional genomics [269] in digestive diseases [270], human gastric cancer organoids [271], and nanoparticles-mediated CRISPR-Cas9 gene therapy in retinal diseases [272]. 

These technologies improve the physiological capacity of experimental models, increase the understanding of the mechanisms associated with the disease, improve the disease modeling, and offer an opportunity for personalized medicine with development of novel therapies [273,274]. Hence, lentiviral transgenesis, CRISPR/Cas9 gene editing, and single-cell readouts were established to study genes function and bring out genetic alterations in organoids. However, the dynamic environment of organoids and their level of complexity also present challenges for genetic editions [275,276].

## 12. Immunotherapy in Organoids 

One area of research that attracted attention was the use of organoids to study the response of cancer cells to immunotherapy. Immunotherapy is a type of treatment that harnesses the power of the immune system to fight cancer [277]. 

A better understanding of the immunological answer of tissues can be gained, enabling the exploitation of immunity in physiological conditions of cancer and a potential capacity for personalized immunotherapy. The co-culture, microfluidic cultures, and ALI organoid systems can resemble the tumor microenvironment. They also provided more profound insights into the interactions between epithelial-immune cells established on their endogenous allocation and immune cells (B, T, and natural killer cells and macrophages) [278,279,280]. Immunotherapy in 3D tumor models became quite promising due to its success in cancer treatment. It was speculated that clinical trials may be carried out to obtain treatment for different types of cancer [87,132].

Tumor organoids derived from the patient’s tumor tissue are a suitable model for epithelial cancers. Tumor organoids can be effectively expanded, cultured for an extended period, and cryopreserved, enabling the creation of biobanks and high-throughput drug screening. Furthermore, they are less expensive than animal models and have fewer associated ethical issues, which led to their use as pre-clinical models to test anticancer agents. In vitro models of organs are being developed using patient-specific iPSCs for a personalized immunotherapy study, identification of new combinations of therapies, and characterization of side effects [277,280,281]. 

Depending on the type of patient-derived tumor organoids, protocols and culture conditions vary, generally with a higher success rate than for cell lines. Tumor organoids were already successfully established for different types of tumors, including brain, breast, kidney, and liver, and allow the propagation of prostate tumors, less aggressive pancreas, and low-grade gliomas. The use of organoids as models of patient-derived glioblastoma was shown better to recapitulate the genetic and phenotypic characteristics of glioblastoma. Additionally, it could bring new immunotherapeutic approaches against aggressive brain cancer [282]. In appendix cancer organoids, immunotherapy showed measurable efficacy [283]. Furthermore, organoids can be applied as a model of rare diseases in clinical trials, given that there is a gap in pre-clinical models and, consequently, a lack of more profound knowledge [283].

Organoids might be a valuable tool in the field of immunotherapy research. They provide a unique opportunity to study cancer in a more physiologically relevant way and to test new immunotherapies in a more accurate representation of the human body. Research in this area is ongoing and has the potential to lead to new and more effective treatments for cancer [279].

## 13. Limitations of Organoids Technology

Organoids are a technology with many potential benefits and applications, but there are also several limitations that must be considered. The generation of organoids has a high production cost, mainly when obtained from iPSCs, because of material price, iPSCs maintenance, long differentiation and maturation time of organoids [284], depending on protocol differentiation and specificity (vascularization and cell type differentiation) [143,285,286,287,288]. Additionally, the lack of vascularization in organoids limits the supply of oxygen and nutrients compared with in vivo cells near capillaries [145,150,289]. The blood vessels are not simple structures, as they vary depending on the size of the blood vessels (wide or capillaries), the layer, and the different types of cells by which they are constituted, increasing the complexity of these structures. Although human blood vessels can be grown in vitro [290], organoid vascularization is very challenging to establish [289,291,292]. 

Thus, without vascularization, there is no adequate perfusion of the medium throughout the organoid, leading to necrosis and apoptosis of some cells, especially the most internal ones, as the supply of oxygen and nutrients is insufficient. These limitations are disadvantageous in this system model because they compromise the development capability or maturation [142,145,293]. For example, in cerebral organoids, the vascularization of the SVZ region is necessary in the later stages of brain development for neural progenitor cell differentiation and maturation. Therefore, without vascularization, cerebral organoids obtained a reduced progenitor population and absence of correct cortical plaque formation [150]. However, cerebral organoids transplanted in vivo mostly performed complete maturation [294]. One way of minimizing this limitation is using bioreactors or microfluidics to help the perfusion medium and deliver nutrients to the organoids, leading to more prominent and mature organoids [150,295]. 

Thus, organoids are not developed as a full-sized organ, and they do not have the same level of complexity or function being limited to replicate the same physiological processes as a full-sized organ. Moreover, to generate and maintain organoids, a specific set of conditions are required, such as a specific type of cell culture, to grow and survive. The interactions between different cell types are essential for resemble in vivo conditions and organs. As an example, the interaction between neural and non-neural cells, such as microglia, hematopoietic cells, and meninges, with yolk sac-derived macrophages that generate differentiated microglia and pericytes from neural crest cells are necessary for neural and vascular system development [296,297]. Therefore, integrating non-neural cells into the cerebral organoid model, such as astrocytes, oligodendrocytes, microglia, meningeal cells, or vascular cells, is necessary [150]. The lack of non-neural cells in the cerebral organoids, mainly because of neuroectodermal fate in the embryoid body, implies limitations in mimicking neurological disorders and understanding pathophysiological alterations, such as in strokes [298].

In the future, establishing a circulatory system for cerebral organoids is essential to perform more detailed studies about neurological diseases. Moreover, a lack of gyrification in the human neocortex cerebral organoids is common in most current protocols and is necessary for a deeper understanding of the molecular and cellular roots of neocortical expansion and folding [167]. 

One challenges is the replication of results between different laboratories; this may limit the ability to use organoids in large-scale studies. Furthermore, the variability of organoids in batch-to-batch production is caused by stem cell organization mechanisms, the cell types of formation control absence, the variability of position of the organ regions, and the organoid size and shape difference. Additionally, the differentiation and growth factors available in the culture system limit not only the unbiased drug screening studies but also the modeling diseases in quantitative studies and the standardization of scientific research [143,144,150,284,299]. Hence, the organoids technology is valuable for studying human disorders, although there are restrictions that must be overcome, such as low homogeneity and high costs [21,145]. 

In addition, organoid reproducibility and stability are essential for beneficial therapeutic screening, such as the scRNA-seq method, which allows transcriptomic expression analysis and molecular characterization. Therefore, bioengineering techniques were in development to improve organoids’ controllability, reproducibility, size, shape, and composition. Microfluidic organ-on-a-chip technology, microwell arrays, and droplet-based microfluidics are some of the new approach methods holding great potential [167].

## 14. Conclusions

The use of culture systems significantly evolved in recent decades. In this article, we presented current knowledge about different cell culture systems, including their advantages and disadvantages.

This review focused on the advantages and limitations of using organoid systems as in vitro models. Organoid technology accelerated the potential of culture systems to model diseases in vitro and removed the ethical constraints associated with the use of animals. Organoids have immense potential in tissue biology research, disease modeling (including genetic, infectious, and malignant diseases), and alternative cell-based therapy, as well as the possibility of reducing the use of animal models and associated ethical issues. Nevertheless, as previously discussed, these models still require improvements before their full potential can be realized. 

Despite the organoid technology potential, it needs to be closely monitored. Standardization in the culture process and achieving a level of production at scale is fundamental and necessary for tissue engineering to allow the production of tissues and organs to be used in transplants without ethical concerns. 

The use of organoids grew and evolved significantly in recent years, among different areas of the scientific community, and their potential and versatility are undeniable, despite their limitations. Future perspectives on organoid technology’s application in the research were explored in detail in this review. This 3D culture technology allows the modeling of different diseases, with potential applicability in regenerative medicine, gene editing, and immunotherapy, leading to the study of more effective targeted therapies.

## Figures and Tables

**Figure 1 cells-12-00930-f001:**
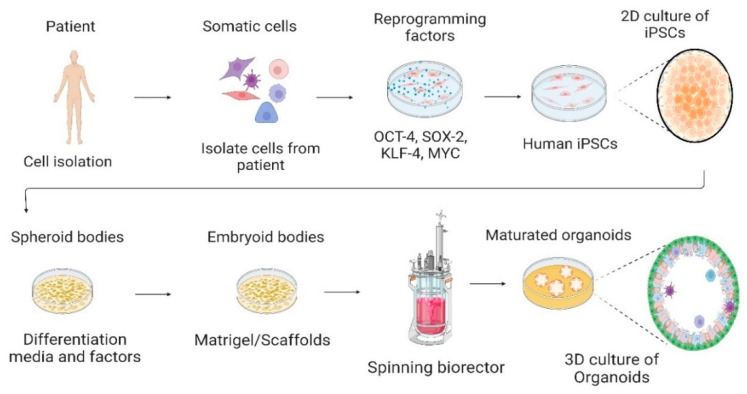
Culture of human-induced pluripotent stem cells in the 2D culture system and differentiation into 3D organoids. Somatic cells are isolated from a human patient, cultured in vitro, and transduced by pluripotency transcription factors, such as OCT-4, SOX-2, KLF-4, and c-MYC. Through reprogramming that induces a pluripotent state in somatic cells, induced pluripotent stem cells (iPSCs) are generated. Cells are cultured and differentiated under specific factors and matrices to generate organoids. Created with BioRender.com, accessed on 23 July 2022.

**Figure 2 cells-12-00930-f002:**
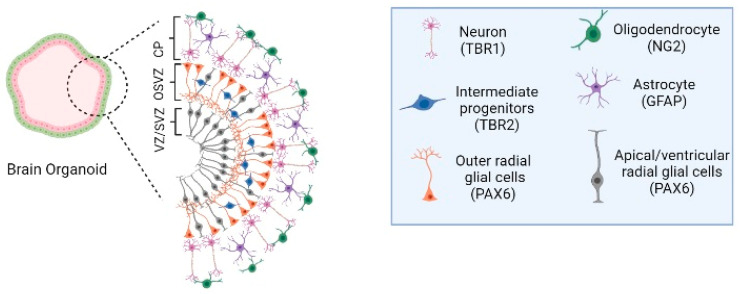
Human cerebral organoids and the timeline of human brain development. (**left**)—Cellular and cytoarchitectural organization of human cerebral organoids. The human cerebral organoid models can be composed of cell types that are associated with the subventricular zone (SVZ), an outer SVZ (OSVZ), and the cortical plate (CP) of the human brain. SVZ region is constituted by apical radial glia cells and ventricular radial glial cells (PAX6 immunohistochemistry marker), the OSVZ structure by outer radial glial cells (PAX6 immunohistochemistry marker) and intermediate progenitors (TBR2 immunohistochemistry marker). The CP is composed of diverse neuronal subtypes, depending on the CP localization, into deep and upper layers, such as mature neurons (MAP2/TBR1 immunohistochemistry marker), early neurons (TUJ1 immunohistochemistry marker), oligodendrocytes (NG2 immunohistochemistry marker), and astrocytes (GFAP immunohistochemistry marker). Image created with BioRender.com, accessed on 23 July 2022.

**Figure 3 cells-12-00930-f003:**
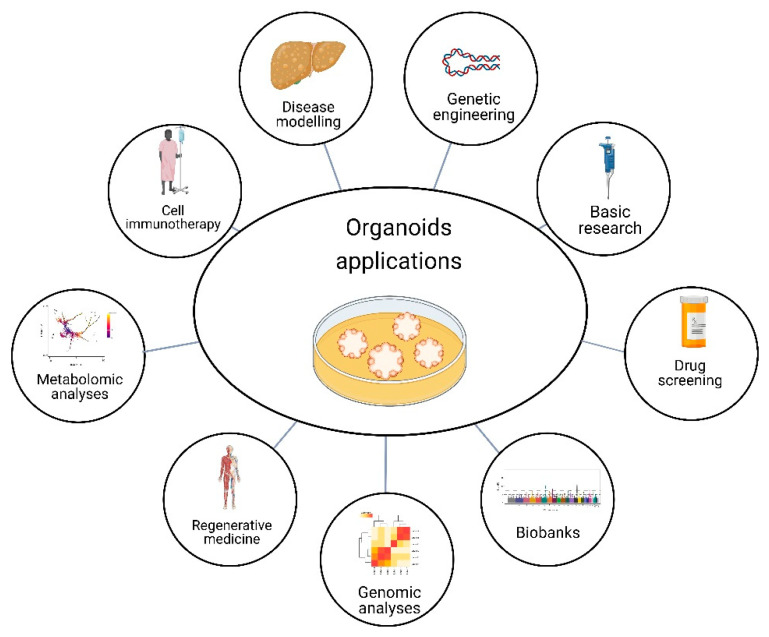
Organoid applications. Organoids have the potential for several purposes, such as basic research, drug screening, genomic and metabolomic analysis, and gene editing to explore disease-linked alleles, called genetic engineering. Moreover, organoids can model diseases, have applications in cell immunotherapy and regenerative medicine, and provide the opportunity to have biobanks. Created with BioRender.com, accessed on 23 July 2022.

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
