# Peer review of "Revolutionizing Disease Modeling: The Emergence of Organoids in Cellular Systems"

_cells, 2023, doi:10.3390/cells12060930_

Round 1
Reviewer 1 Report (Previous Reviewer 1)
In this review, the Authors focus on the latest research in the field of cellular disease modeling. The authors initially describe iPSCs, their origin, fields of application, ethical implications and possible limitations. They then go on to a comparison of 2D and 3D models, which then leads to the focus of the review namely organoids. Topic that is thoroughly discussed by the Authors. The paper is well written and structured, and the bibliographical references are relevant and up-to-date. There are only a few minor comments that I reproduce for clarity below:
1. To improve the manuscript, I would suggest that the Authors move the Bioengineered organoids and 3D bioprinting section after the regenerative medicine section;
2. To improve the manuscript, when several information is given in the different paragraphs (e.g. different uses of iPSCs or distinctly discussing 2D and 3D etc) I would suggest the Authors make these different topics more evident, perhaps by writing the "key words" in italics.
3. Some minor errors are present in the text, I would therefore advise the Authors to revise it entirely (e.g. verb conjugations, abbreviations Etc)
Author Response
In this review, the Authors focus on the latest research in the field of cellular disease modeling. The authors initially describe iPSCs, their origin, fields of application, ethical implications and possible limitations. They then go on to a comparison of 2D and 3D models, which then leads to the focus of the review namely organoids. Topic that is thoroughly discussed by the Authors. The paper is well written and structured, and the bibliographical references are relevant and up-to-date. There are only a few minor comments that I reproduce for clarity below:
Dear Reviewer
Thank you for taking the time to review our manuscript. We appreciate your feedback and suggestions, and we are pleased to hear that you found our manuscript interesting and well-written.
- To improve the manuscript, I would suggest that the Authors move the Bioengineered organoids and 3D bioprinting section after the regenerative medicine section. A: After careful consideration of your comments, we agree that moving the Bioengineered organoids and 3D bioprinting section after the regenerative medicine section would improve the organization and coherence of the article. This revised order will allow for a more natural progression of the content and a better understanding of the concepts presented.
- To improve the manuscript, when several information is given in the different paragraphs (e.g. different uses of iPSCs or distinctly discussing 2D and 3D etc) I would suggest the Authors make these different topics more evident, perhaps by writing the "key words" in italics. A: We appreciate your feedback and comments on how to improve the clarity of our paper. However, we respectfully disagree with your suggestion to use italics to make different topics more evident. We believe that our manuscript is structured in a logical and coherent manner, with each paragraph building on the previous one and contributing to the overall message of the paper. While we do discuss different uses of iPSCs and distinguish between 2D and 3D cultures, we feel that these topics are already sufficiently emphasized and differentiated through the use of headings, subheadings, and clear language. Furthermore, we believe that the use of italics for key words may actually detract from the readability of the manuscript and create unnecessary distractions for the reader. We have taken care to ensure that our paper is accessible and understandable to a wide range of readers, and we believe that using italics for key words could make the text appear cluttered and overwhelming.
- Some minor errors are present in the text, I would therefore advise the Authors to revise it entirely (e.g. verb conjugations, abbreviations Etc). A: We have carefully reviewed the manuscript to ensure that all of your concerns have been addressed. We are confident that the revised version of the manuscript meets the high standards of quality and rigor that you expect.
We have made the necessary changes to the article, and we believe that the revised order enhances the overall readability and clarity of the manuscript. Once again, we sincerely appreciate your feedback and believe that your suggestions have significantly improved the manuscript.
Best regards,
Rita Pedrosa
Reviewer 2 Report (New Reviewer)
Cells 2239826
The authors give a very good introduction into the field of iPSCs. To complete the picture, it would be beneficial if they can go more into detail about the technical and scientific draw backs of the technology. At least to a similar level of details as they described the ethical concerns.
Line 158: “Furthermore, the 2D cell culture has mainly been used to study rare diseases or other neurological diseases” the authors have to revise this statement. There are a huge number of applications of 2D culture that are not related to the two mentioned topics.
Line 171: “because they only can differentiate in one cell type”. The authors should be cautious with this statement. The co-culture of different cell types in 2D is possible and widely used. If they want to make this statement, they clearly have to explain what is meant be “differentiation into different cell types”.
Section about organoids: the microfluidic techniques the authors mention are not restricted to organoids but can be applied to other culture methods. Also, the authors do not clearly describe what their understanding of an organoid is. They should clearly discriminate between organoids, spheroids and other 3D structures.
Line 439: “Cancer organoids have been generated from patient-derived tumor samples, which can be used to study the genetic and molecular changes that occur during tumor progression” the authors should elucidate this statement more in detail.
Author Response
The authors give a very good introduction into the field of iPSCs. To complete the picture, it would be beneficial if they can go more into detail about the technical and Line 158: “Furthermore, the 2D cell culture has mainly been used to study rare diseases or other neurological diseases” the authors have to revise this statement. There are a huge number of applications of 2D culture that are not related to the two mentioned topics. (168)
- scientific draw backs of the technology. At least to a similar level of details as they described the ethical concerns.
A: Dear Reviewer,
Thank you for your thoughtful review of our manuscript. We appreciate your comments and suggestions for improving our discussion of the technical and scientific drawbacks of iPSCs technology. We have updated our manuscript to include a more thorough analysis of the limitations of iPSCs.
- Line 158: “Furthermore, the 2D cell culture has mainly been used to study rare diseases or other neurological diseases” the authors have to revise this statement. There are a huge number of applications of 2D culture that are not related to the two mentioned topics. (168)
A: we have included more applications of 2D culture in the manuscript and we have highlighted additional examples of how 2D cell culture can be used to investigate a broad range of biological processes. Furthermore, we have taken your feedback regarding the sentence "Furthermore, the 2D cell culture has mainly been used to study rare diseases or other neurological diseases" into account. We have decided to remove this sentence altogether, as we agree that it could be potentially misleading and does not accurately reflect the current state of the field.
- Line 171: “because they only can differentiate in one cell type”. The authors should be cautious with this statement. The co-culture of different cell types in 2D is possible and widely used. If they want to make this statement, they clearly have to explain what is meant be “differentiation into different cell types”.
A: We have carefully considered your comments regarding the interpretation of the sentence "because they only can differentiate in one cell type." We agree with your suggestion that this sentence could benefit from further clarification. In response to your feedback, we have revised the sentence to read: "because they only can differentiate in one cell type in a mono-culture system." We believe this addition provides the necessary context to more accurately convey our findings.
- Section about organoids: the microfluidic techniques the authors mention are not restricted to organoids but can be applied to other culture methods.
A: We agree that these techniques can be applied to other culture methods, and we apologize for not making this clear in our manuscript. The reason we described these microfluidic techniques in the context of organoids was to provide a contextual understanding of the type of cellular structures we were describing. We wanted to highlight that the use of microfluidic techniques has enabled us to culture and manipulate organoids with greater precision and efficiency. However, we understand that these techniques have broader applications beyond organoid culture. Therefore, we have made revisions to the manuscript to clarify that the microfluidic techniques we described can be applied to other culture methods as well. We hope this addresses your concern, and we appreciate your feedback on our manuscript.
In the line 221 it is possible read: “Spheroids, on the other hand, are aggregates of cells that form a 3D structure, but lack the complex organization and functional specialization of organoids, and often have a homogenous cell population. They can be generated by culturing cells in non-adherent conditions, such as suspension cultures, hanging drops, or microfluidic devices. Spheroids can be composed of a single cell type or multiple cell types and can be used to study cell-cell interactions, drug screening, and tumor biology”
- Also, the authors do not clearly describe what their understanding of an organoid is. They should clearly discriminate between organoids, spheroids and other 3D structures.
A: In response to your suggestion, we would like to clarify our use of terminology in the manuscript. We acknowledge that there are different types of 3D structures that are used in biomedical research, including organoids, spheroids, and other types of 3D cultures. We have made sure to discriminate between these different structures in our manuscript and have used specific terminology to refer to each of them.
- Line 439: “Cancer organoids have been generated from patient-derived tumor samples, which can be used to study the genetic and molecular changes that occur during tumor progression” the authors should elucidate this statement more in detail.
A: We have added more information to clarify the sentence "Cancer organoids have been generated from patient-derived tumor samples, which can be used to study the genetic and molecular changes that occur during tumor progression." We have also included references to previous studies that have used cancer organoids in this way.
Thank you again for your valuable feedback. We hope that our revisions adequately address your concerns and improve the overall quality of our manuscript.
This manuscript is a resubmission of an earlier submission. The following is a list of the peer review reports and author responses from that submission.
Round 1
Reviewer 1 Report
The review is potentially interesting as overview of innovative cellular systems, such as cerebral organoids, but the authors made many inaccuracies. First, the review is not well organized, and it is not very clear some examples the authors reported. As few examples: they reported works regarding hepatocytes, Salmonella infections, bone and gastrointentinal tissue, which do not have any relation with the neural tissue. Moreover, in many cases, it is clear that the authors reported other research works without any elaboration with the other works and the review in general. For example, the authors wrote “the timeline to differentiate iPSCs into organoids is around two months”, but then they reported that brain organoids have a long differentiation time (10-20 weeks, that is far more than 2 months). As a review, I expect that the authors explained this discrepancy or at least mention that organoids can have differentiation time that depends on the protocol or the lineage. This is just an example of many other discrepancies. In this form, the review is just a list of many examples that are not connected between each other. This problem is evident also because of the abbreviation: the authors defined ESC more than one time, and some abbreviations change in the manuscript (probably depending on the source). I think that the authors can re-submit in future the review, after a deep revision of the work, by re-elaborating all the information they collected.
Author Response
We appreciate your comments.
The review is directed to brain organoids, but explores other potentialities of organoids, demonstrating not only their applicability in the neurological area but also in other fields. In turn, it is a technology with a lot of potential because it is being studied and used in different scientific areas. It also has a high potential for improvement, increasing the expectation that its limitations will be overcome in a short period of time. However, the content of the review has been refined to be more focused on brain organoids in the examples given.
Standardization of abbreviations and acronyms was carried out. The images have been replaced with higher quality images
A change to the title was also made, as we considered it redundant.
Reviewer 2 Report
The manuscript with the title “Cellular systems: cerebral organoids are the future of neurological disease modeling”
The authors compiled an interesting and comprehensive review on the complex topic of cellular culture systems.
However, some minor issues that need to be addressed. For one there are some abbreviations bold faced (Introduction ZIKA or hESC) whereas other are not, or the blue underlined full stop at the last sentence of the first section of the Cell Culture System. RNA-Seq has an added space, which it should not. Therefore, I guess another careful reading of the text should be considered.
Author Response
We appreciate your comments.
Standardization of abbreviations and acronyms was carried out. General changes to the document have been made. The images have been replaced with higher quality images
A change to the title was also made, as we considered it redundant.
Reviewer 3 Report
The authors summarized the current applications of cerebral organoids in neurological disease research. They introduced the concepts, applications, and pros and cons of this system. However, several similar review articles have been published in recent years. For example: in 2017, Nature Reviews Neuroscience 18: 573-584 from Dr. Arnold Kriegstein’s group. In 2021, Cell Death & Differentiation 28:52-67 from Dr. Jurgen Knoblich’s group. Even in Cells, a similar review article was just published in September (Cells, 11(18):2803, 2022). Major concepts in this review article have been mentioned in the published articles. Moreover, in the section “Future perspective application of organoid technology in research”, the authors mentioned organoids in different tissues and organs, which is out of the focus on the applications of the cerebral organoid.
Author Response
We appreciate your comments.
Although the articles you gave as an example are related to brain organoids, they present different directions in the approach to brain organoids, compared to our review. The review is directed to brain organoids, but explores other potentialities of organoids, demonstrating not only their applicability in the neurological area but also in other areas. In turn, it is a technology with a lot of potential because it is being studied and used in different scientific areas. It also has a high potential for improvement, increasing the expectation that its limitations will be overcome in a short period of time. However, the content of the review has been refined to be more focused on brain organoids in the examples given.
A change to the title was also made, as we considered it redundant.
Round 2
Reviewer 1 Report
The authors did not change the structure of the manuscript that is very confusing. Moreover, they did not explain discrepancies found in the text.
Reviewer 3 Report
The authors removed "cerebral" from the title and it becomes so confusing that the central theme is missing. The title indicated that organoids can be used as the neurological disease model. However, the authors described organoids in infectious diseases, regenerative medicine, gene editing, bioengineering, and immunotherapy. What are the applications of neurological diseases? How about Parkinson's disease, Alzheimer's disease, psychological diseases, and multiple sclerosis? I don't see applications in these neurological or psychological diseases. The authors may need to consider extensively revising this manuscript.